# MOCHA: Multi-sample Omics Cohorts with Human Annotation

## Abstract

In spatially resolved transcriptomics (SRT) research, gene expression profiling with spatial context has enabled spatial domain identification within single tissue samples. Extending these analyses to multiple biological samples presents additional challenges, including cross-sample variability and batch effects. Method development has been limited by the lack of datasets that combine multi-subject cohorts with expert-derived annotations. We present MOCHA (Multi-sample Omics Cohorts with Human Annotation), a curated resource for developing and evaluating multi-sample SRT methods. MOCHA integrates molecular profiles, spatial profiles, and high-resolution Hematoxylin and Eosin (H&E) images across multiple subjects, with each sample paired with domain annotations from expert pathologists. For algorithm development and evaluation, MOCHA provides standardized data organization, efficient storage formats for large-scale processing, and protocols for handling batch effects in multi-sample integration.

## 1 Introduction

Spatially resolved transcriptomics (SRT) links gene expression profiles to precise tissue coordinates, enabling quantitative analysis of microanatomy and cellular organization at high resolution. Multiple platforms now make SRT broadly accessible, including sequencing-based assays such as 10x Genomics Visium and Slide-seq (Ståhl et al., 2016; Tian et al., 2023) and imaging-based assays such as MERFISH (Chen et al., 2015) and STARmap (Wang et al., 2018). The resolution of these technologies varies from multi-cellular spots to near single-cell measurements, but all require computational approaches that can identify coherent tissue domains by combining molecular profiles with spatial information.

Several repositories have been developed to organize publicly available datasets, including SORC for cancer research (Zhou et al., 2024), Aquila for cross-disease analyses (Zheng et al., 2023), and others such as SODB (Yuan et al., 2023), STOmicsDB (Xu et al., 2022), and SpatialDB (Fan et al., 2020). Despite this progress, multi-subject datasets with expert-generated spatial annotations remain limited. This gap constrains systematic method development for multi-sample integration—an essential setting for cohort-level studies that must model biological heterogeneity alongside technical variation.

Methodological advances underscore this need. Early work emphasized single-sample domain identification, including Bayesian modeling approaches such as BayesSpace (Zhao et al., 2021) and deep learning methods that integrate histology, including iIMPACT (Jiang et al., 2024). More recent approaches—such as BASS (Li & Zhou, 2022), BayeSmart (Guo et al., 2024), and graph-based methods like STAGATE (Dong & Zhang, 2022)—extend analysis to multiple samples using distinct strategies, from clustering across tissues to learning shared representations. Additional challenges, such as deconvolution of mixed spots (Chen et al., 2022; 2023; Luo et al., 2024) and correction for batch effects, reinforce the importance of datasets that provide aligned molecular, spatial, and histological information together with expert annotations.

We introduce MOCHA, a Multi-sample Omics Cohorts with Human Annotation database for training and evaluation of multi-sample SRT methods. MOCHA aggregates multi-subject datasets that each include a gene expression matrix, spatial coordinates, and a co-registered high-resolution Hematoxylin and Eosin (H&E) image (Chan, 2014). Each sample is accompanied by spatial domain

labels produced by an expert pathologist, enabling evaluation of domain delineation and representation learning in multi-sample contexts. To promote reproducibility and accessibility, MOCHA is released in formats readily usable with Python and R and distributed for integration into existing pipelines.

## 2 DATASETS

To assemble a resource for multi-sample spatial domain identification, we curated a set of publicly available SRT datasets. Following an approach similar to that in the STimage-1K4M review Chen et al. (2024), we systematically searched repositories including 10x Genomics, Gene Expression Omnibus (GEO), and Spatial Research. Our selection criteria required each study to provide a cell-by-gene expression count matrix, a spatial coordinate matrix, and cellular annotations delineated by a pathologist using the corresponding H&E images.

This search yielded 10 distinct cohorts, summarized in Table 1. Cancer-related datasets include HER2-positive breast cancer (BC_HER2+) (Andersson et al., 2021), high-plasticity subtypes (BC_HP) (Coutant & et al., 2023), recurrent neoplastic heterogeneity (BC_NP) (Wu et al., 2021), triple-negative breast cancer (BC_TNBC) (Wang et al., 2024), colorectal cancer consensus molecular subtypes (CRC_CMS) (Valdeolivas et al., 2024), kidney cancer with tertiary lymphoid structures (KC_TLS) (Dawo et al., 2023), lung cancer with tertiary lymphoid structures (LC_TLS) (Dawo et al., 2023), and renal cell carcinoma with tertiary lymphoid structures (RCC_TLS) (Meylan & et al., 2022), along with human dorsolateral prefrontal cortex (DLPFC) (Maynard et al., 2021) and mouse olfactory bulb (MOB) (Ståhl & et al., 2016).

Table 1: A summary of the SRT datasets. (BC: Breast cancer; CRC: Colorectal cancer; DLPFC: Dorsolateral prefrontal cortex; KC: Kidney cancer; LC: Lung cancer; MOB: Mouse olfactory bulb; RCC: Renal cell carcinoma)

| Cohort | Tissue | Technology | Subjects | Samples |
|---|---|---|---|---|
| BC_HER2+_10x | HER2-positive (HER2+) breast cancer | 10x Visium | 8 | 8 |
| BC_HP_10x | High-plasticity (HP) breast cancer subtypes | 10x Visium | 12 | 14 |
| BC_NP_10x | Recurrent neoplastic (NP) cell heterogeneity in breast cancer | 10x Visium | 6 | 6 |
| BC_TNBC_ST | Triple-negative breast cancer (TNBC) | ST | 94 | 94 |
| CRC_CMS_10x | Colorectal cancer consensus molecular subtypes (CMS) | 10x Visium | 11 | 14 |
| DLPFC_10x | Dorsolateral prefrontal cortex | 10x Visium | 3 | 12 |
| KC_TLS_10x | Kidney cancer with tertiary lymphoid structures (TLS) | 10x Visium | 3 | 3 |
| LC_TLS_10x | Lung cancer with tertiary lymphoid structures (TLS) | 10x Visium | 5 | 5 |
| MOB_ST | Mouse olfactory bulb | ST | 1 | 12 |
| RCC_TLS_10x | Tertiary lymphoid structures (TLS) in renal cell carcinoma | 10x Visium | 23 | 23 |

These cohorts span a wide range of tissue types and disease contexts, encompassing both human and mouse studies, and multiple technological platforms (10x Genomics Visium and ST). The scale also varies substantially, with BC_TNBC including 94 subjects and 94 samples, while smaller datasets such as KC_TLS and LC_TLS consist of only three and five samples, respectively. Together, these datasets enable evaluation of multi-sample spatial domain identification methods. A summary of molecular characteristics, including the number of spots, genes, and data sparsity for each cohort, is presented in Figure 1.

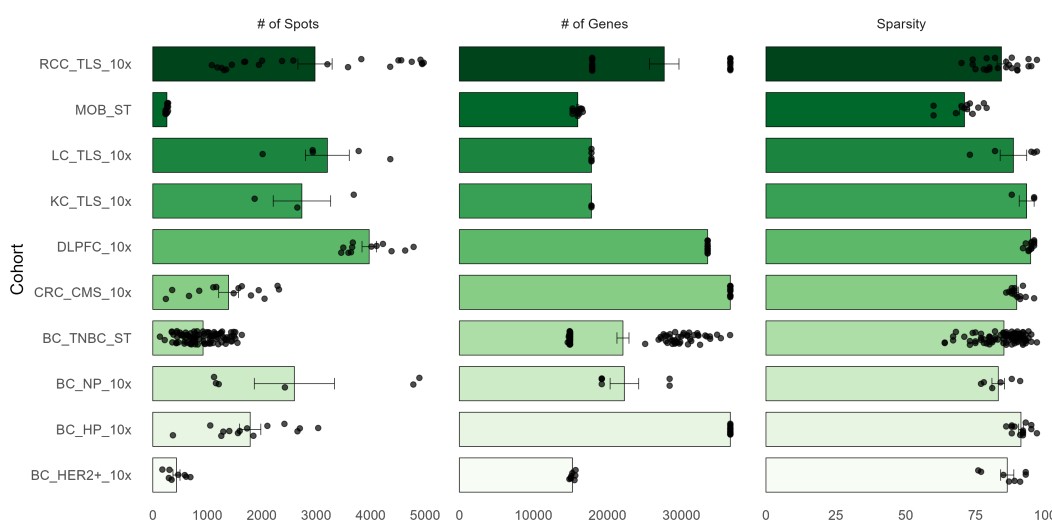

Figure 1: A summary of the molecular profiles for each cohort.

## 3  PRE-PROCESSING AND BATCH EFFECT CORRECTION

A standard pipeline for preprocessing multi-sample SRT data starts by concatenating the raw gene expression matrices from each sample over a set of common genes, followed by library size normalization to correct for variability in sequencing depth. This adjustment can be performed using packages such as `scater` and `scran`, which implement techniques such as the trimmed mean of M-values (TMM), relative log expression (RLE), and upper-quartile scaling (Robinson & Oshlack, 2010; Anders & Huber, 2010; Bullard et al., 2010; McCarthy et al., 2017). Alternatively, frameworks such as `Seurat` and `scanpy` apply a global-scaling approach in which counts for each cell are divided by the total count, rescaled to a fixed scaling factor (e.g., 10,000), and log-transformed to stabilize variance (Hao et al., 2023; Wolf et al., 2018).

Following normalization, dimensionality reduction can be performed through feature selection or projection methods. Feature selection can involve identifying spatially variable genes (SVGs) using methods such as SPARK-X (Zhu et al., 2021; Zhao et al., 2021; Jiang et al., 2024), or highly variable genes (HVGs), which are generally preferred in studies involving multiple subjects to reduce inter-subject variability (Li & Zhou, 2022). Dimensionality reduction can also be achieved by projecting the data into a lower-dimensional space using techniques such as PCA, t-SNE, UMAP, or graph attention autoencoders as implemented in STAGATE (van der Maaten & Hinton, 2008; Becht et al., 2019; Dong & Zhang, 2022).

Batch correction can be subsequently performed to adjust for systematic variation between samples. One common approach is to operate on reduced feature spaces using techniques such as Harmony (Korsunsky et al., 2019; Li & Zhou, 2022; Guo et al., 2024). An overview of this batch effect correction, and feature selection, workflow is demonstrated in Figure 2.

An alternative pipeline for batch correction is implemented in Crescendo, which avoids transformation to a reduced-dimensional space and instead models the raw, integer-valued counts directly (Millard et al., 2025). This approach employs a generalized linear mixed model (GLMM) in which the batch is included as a random effect, preserving the discrete structure of the data. Crescendo can extend single-sample spatial clustering models such as BayesCafe, which relies on the zero-inflated negative binomial (ZINB) distribution (Li et al., 2024), to multi-sample settings by integrating batch correction directly within the generative hierarchy.

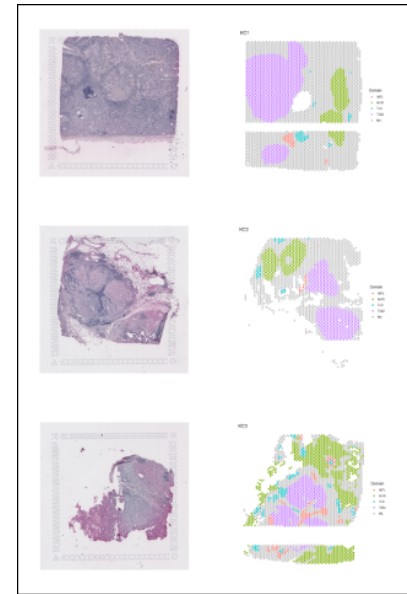
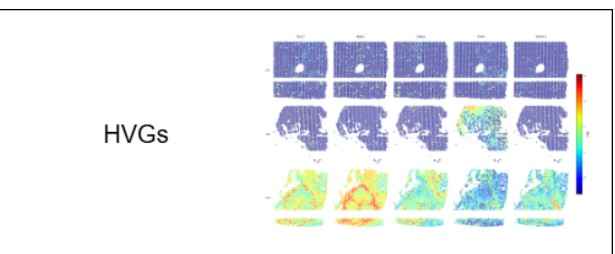
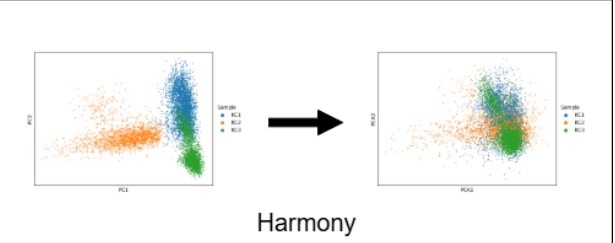

Figure 2: AA standard pipeline for feature selection with HVGs and batch effect correction using Harmony, illustrated with the KC_TLS_10x cohort (Dawo et al., 2023).

# 4    MULTI-SAMPLE SPATIAL CLUSTERING METHODS

Recent advances in computational modeling have led to methods that extend spatial transcriptomics analysis from single-sample to multi-sample settings. These approaches are designed to integrate spatial and molecular information across subjects while accounting for technical and biological variability.

As summarized in Table 2, BayeSMART is a Bayesian framework for multi-sample spatial clustering that integrates reconstructed single-cell information from histology images with spatial gene expression (Guo et al., 2024). BASS is a hierarchical Bayesian model that jointly performs cell type clustering and spatial domain identification across samples (Li & Zhou, 2022). STAGATE is a graph attention autoencoder that generates low-dimensional embeddings by combining spatial neighborhood structure with molecular profiles (Dong & Zhang, 2022).

Table 2: A summary of the existing multi-sample spatial clustering methods. These Bayesian (Bayes) or deep learning (DL) approaches use Principal Component Analysis (PCA) or autoencoders (AE) for dimension reduction. Additionally, BayeSMART integrates information from H&E images.

| Method | Dimension reduction | H&E | Approach | Language | Year |
|---|---|---|---|---|---|
| BayeSMART | PCA | ✓ | Bayes | R/C++ | 2024 |
| BASS | PCA | | Bayes | R/C++ | 2022 |
| STAGATE | AE | | DL | Python | 2022 |

In a majority of the cancer studies included in MOCHA, the detailed pathologist annotations can be grouped into four broad categories: immune, stroma, tumor, and normal. These groupings, described in the Supplementary Material, provide a consistent reference structure for applying multi-sample spatial clustering methods while accommodating variability across cohorts.

AUTHOR CONTRIBUTIONS

ACKNOWLEDGMENTS

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
