# OpenReview forum: "MOCHA: Multi-sample Omics Cohorts with Human Annotation"
_ICLR.cc/2026/Conference — ICLR 2026 Conference Withdrawn Submission_

### Official Review · Reviewer_aMDZ · 2025-10-14

**Soundness:** 3
**Presentation:** 3
**Contribution:** 1
**Rating:** 0
**Confidence:** 5

**Summary:**

This paper introduces MOCHA (Multi-sample Omics Cohorts with Human Annotation), a curated resource for benchmarking and developing multi-sample spatial transcriptomics (SRT) analysis methods. The dataset integrates molecular profiles, spatial coordinates, and co-registered H&E images across ten cohorts from multiple diseases and species, each annotated by expert pathologists. The authors describe data preprocessing, normalization, and batch correction pipelines, and summarize existing multi-sample clustering methods (BASS, BayeSMART, STAGATE). The work aims to provide a standardized foundation for evaluating spatial domain identification and integration methods across biological replicates.

**Strengths:**

The paper addresses an important unmet need in the field: the lack of standardized, well-annotated multi-sample SRT benchmarks. The inclusion of multiple cancer and brain datasets with expert pathologist annotations enhances biological credibility. The organization of data into interoperable formats (Python/R) and detailed preprocessing protocols may facilitate reproducibility and adoption by the community. The authors also provide a clear overview of current computational approaches, positioning MOCHA as a potentially useful testing bed for new methods.

**Weaknesses:**

Despite its relevance, the paper is primarily descriptive and lacks a clear technical or methodological contribution. It does not include quantitative benchmarking or demonstrate the practical impact of MOCHA on existing algorithms. The dataset aggregation appears to rely almost entirely on previously published public data, with limited novelty in curation methodology. The “expert annotation” process is insufficiently detailed — e.g., the number of annotators, inter-rater agreement, and labeling consistency are not reported. The writing also suffers from redundancy and lacks a cohesive narrative connecting the biological motivation with the computational pipeline. Overall, the manuscript feels more like documentation for a dataset release than a scientific study.

**Questions:**

1. There is no demonstration that MOCHA improves method benchmarking, reproducibility, or model generalization compared to existing datasets.
2. The process for expert-generated labels (criteria, consistency checks, number of annotators) should be clearly described.
3. The heavy focus on breast and colorectal cancer reduces generalizability; inclusion of more non-cancer datasets would strengthen the resource.
4. The preprocessing and batch correction pipelines largely follow standard protocols (Harmony, HVG, Seurat/Scanpy), offering little methodological novelty.
5. The paper should include code examples or benchmark tasks to illustrate MOCHA’s practical integration into real-world pipelines.
6. Figures are low-impact and primarily procedural; replacing some with visual summaries of data diversity or annotation quality would improve readability.
7. There is no explicit discussion of licensing, data provenance, or privacy compliance for reused human data.

---

### Official Review · Reviewer_Xti7 · 2025-11-01

**Soundness:** 1
**Presentation:** 1
**Contribution:** 1
**Rating:** 0
**Confidence:** 5

**Summary:**

This paper wants to talk about MOCHA but seems unfinished.

**Strengths:**

N/A

**Weaknesses:**

Unfinised paper.

**Questions:**

N/A

---

### Official Review · Reviewer_fkum · 2025-11-01

**Soundness:** 2
**Presentation:** 2
**Contribution:** 2
**Rating:** 2
**Confidence:** 3

**Summary:**

This paper introduces MOCHA, a curated resource designed to accelerate the development and evaluation of multi-sample spatially resolved transcriptomics (SRT) methods.

**Strengths:**

- The paper clearly identifies and addresses a significant bottleneck for progress in computational spatial biology. While many SRT analysis methods exist, robust benchmarking has been hampered by the lack of multi-sample datasets with expert-annotated ground truth. This paper provides a direct and practical solution to this problem.
- The core contribution is the thoughtful curation and aggregation of these 10 diverse cohorts. The value is in establishing a strict selection criterion (requiring gene expression, spatial coordinates, H&E, and pathologist annotations) and assembling them into a single, cohesive resource spanning multiple cancer types, technologies, and species.
- By including standardized pre-processing and batch correction protocols (e.g., outlining the use of HVGs and Harmony), the authors lower the barrier to entry for researchers and help ensure that future methods can be compared in a more standardized and reproducible manner.

**Weaknesses:**

- The paper's primary contribution is the aggregation and standardization of 10 existing public datasets . It is not entirely clear what the novel "deliverable" is beyond a list of these datasets. The abstract mentions "standardized data organization" and "efficient storage formats", but this is not detailed. What specific standardization or re-formatting was applied that provides value beyond what a user could get from downloading each dataset from its original source? Overall this paper seems to be better suited for a conference with a separate dataset track.
- The paper stops short of providing any baseline benchmark results with the curated datasets. How do the authors proposed to validate the resource's utility in this case?
- The key value addition is the "expert pathologist" annotations. However, these annotations come from 10 different studies, presumably by 10 different experts with different goals. The paper mentions grouping them into "immune, stroma, tumor, and normal". How consistent are these labels, and what was the mapping process?

**Questions:**

- Is MOCHA a single downloadable package (e.g., on Zenodo) containing all 10 processed cohorts? Or is it a code repository (e.g., GitHub) with scripts to download and pre-process each dataset from its original source? The paper would benefit from a Data Availability type section.

---

### Official Review · Reviewer_HMKS · 2025-11-09

**Soundness:** 2
**Presentation:** 1
**Contribution:** 1
**Rating:** 0
**Confidence:** 5

**Summary:**

This work presents MOCHA, a curated spatial transcriptomics (ST) dataset resource for the research community. MOCHA is a curated collection of ten 10x Visium datasets with batch correction. The authors plan to make this resource available.

**Strengths:**

If made available, MOCHA would be a good resource for researchers in the biomedical / AI community to use for method development in ST.

**Weaknesses:**

- Paper is not organized very well with a paper length of 4 pages. A section on spatial clustering is introduced but does not tie in with the rest of the paper, and it is hard to follow.
- No comparison with SKT-142M or HEST-1k in terms of how their dataset is curated and organized differently.
- No technical novelty or any research experiments produced in this paper.

**Questions:**

No questions.

---

### Note · Authors · 2025-11-14

**Comment:**

We appreciate the reviewers’ evaluations. After reviewing the feedback, we have decided to withdraw the paper from the ICLR.

**Withdrawal Confirmation:**

I have read and agree with the venue's withdrawal policy on behalf of myself and my co-authors.